# Synthesis of Boron-Doped Carbon Nanomaterial

**DOI:** 10.3390/ma16051986

**Published:** 2023-02-28

**Authors:** Vladimir V. Chesnokov, Igor P. Prosvirin, Evgeny Yu. Gerasimov, Aleksandra S. Chichkan

**Affiliations:** Boreskov Institute of Catalysis SB RAS, Prospekt Akademika Lavrentieva 5, Novosibirsk 630090, Russia

**Keywords:** synthesis, graphene, doping, boron

## Abstract

A new method for the synthesis of boron-doped carbon nanomaterial (B-carbon nanomaterial) has been developed. First, graphene was synthesized using the template method. Magnesium oxide was used as the template that was dissolved with hydrochloric acid after the graphene deposition on its surface. The specific surface area of the synthesized graphene was equal to 1300 m^2^/g. The suggested method includes the graphene synthesis via the template method, followed by the deposition of an additional graphene layer doped with boron in an autoclave at 650 °C, using a mixture of phenylboronic acid, acetone, and ethanol. After this carbonization procedure, the mass of the graphene sample increased by 70%. The properties of B-carbon nanomaterial were studied using X-ray photoelectron spectroscopy (XPS), high-resolution transmission electron microscopy (HRTEM), Raman spectroscopy, and adsorption-desorption techniques. The deposition of an additional graphene layer doped with boron led to an increase of the graphene layer thickness from 2–4 to 3–8 monolayers, and a decrease of the specific surface area from 1300 to 800 m^2^/g. The boron concentration in B-carbon nanomaterial determined by different physical methods was about 4 wt.%.

## 1. Introduction

Graphene attracted the attention of researchers due to its exceptional structural, mechanical, and electronic properties [1,2,3,4]. The applications of graphene continue to grow due to its record thermal conductivity, excellent mechanical strength [5,6,7], remarkable ability to carry charges, outstanding thermal stability, and high surface area [8,9,10]. The applications of graphene-based nanoadsorbents, including graphenes, graphene oxides, reduced graphene oxides, and their nanocomposites in water purification were summarized in a recent review [11]. 

The properties of graphene-based materials can be improved by doping with other “I have checked”.elements [12]. Boron, nitrogen, and phosphorous are the main elements used for doping graphene and graphene oxide [13,14]. Graphene doping with boron results in p-type conductivity, whereas phosphorus induces n-type conductivity. Doping with boron and phosphorous increases the surface area and concentration of defects in graphene materials. New applications for using graphene and graphene quantum dots doped with boron or phosphorous as sensors, sorbents, photocatalysts, and electrocatalysts for the detection and reduction of various pollutants were presented in a review [15].

The doping of carbon nanomaterials with boron atoms can alter their electronic properties. Recently, it was found that boron-doped carbon nanomaterials have remarkable properties in electrocatalytic oxygen reduction reactions (ORRs) [16,17,18,19,20]. Based on the obtained data, many researchers believe that B-doped carbon nanotubes and graphene can act as a substitute for expensive platinum catalysts in fuel cells. 

In this article [21], ordered mesoporous carbon doped with boron was prepared by coimpregnation of SBA-15 silica with sucrose and 4-hydroxyphenylboric acid followed by carbonization. The authors found that mesoporous carbon doped with boron had a highly-ordered mesoporous structure, uniform pore size distribution and high surface area. Ordered mesoporous carbon doped with boron can be used as a potentially efficient and cheap metal-free catalyst for ORRs with good stability in an alkaline solution. The boron concentration is the key factor in determining the catalytic activity in ORRs. The high surface area of ordered mesoporous carbon doped with boron makes active sites more accessible in ORRs. 

There are several reports in the literature on the synthesis of boron-doped graphene. In these studies, the conventional CVD method was used for growing graphene on different polycrystalline foils. It was reported that boron-doped graphene could be synthesized using phenylboronic acid as a source of carbon and boron [22]. The effects of the phenylboronic acid mass and graphene growth time on the properties of graphene growing on polycrystalline copper foil in a three-zone CVD system were studied. The nanomaterial prepared by the substitution doping of graphene with boron has high stability, which is particularly important for applications in semiconductor technology, particularly in optoelectronics. However, this method has significant drawbacks. Boron atoms can penetrate both the graphene structure and the structure of polycrystalline foil. Therefore, subsequently, it is difficult to distinguish places where boron atoms are localized. In addition, it is difficult to scale up this method and synthesize uniform graphene films.

In the article [23], graphene oxide prepared by the Hammer’s method was used as the precursor for the synthesis of B-graphene. Boron atoms were successfully introduced into the graphene oxide structure with the concentration of 1.64–1.89 at.%. Boron was introduced into the graphene oxide from boric acid by hydrothermal reduction at 250–300 °C. The presence of B–O, B–C and C–O bonds was confirmed by FTIR spectral analysis. The main drawback of this method for the synthesis of B-graphene is the high concentration of oxygen atoms in the graphene oxide precursor that affects the quality of the final product. The introduction of an additional graphene oxide reduction stage also does not solve this problem because it leads to the formation of many defects in the graphene structure. 

The effect of oxygen and boron co-introduction in porous carbon on the activity of ORRs was studied in [24]. The synthesis of porous carbon doped with oxygen and boron was based on simple carbonization with CO_2_ using NaBH_4_ as a reducing agent, followed by thermal treatment. It was demonstrated that the presence of O–B–C fragments increased the activity of porous carbon in ORRs, and led to high stability in cycles. 

In this study, graphene obtained via the template synthesis [25,26,27] was used as the precursor for the synthesis of B-carbon nanomaterial. The goals of this study were to deposit an additional boron-doped graphene layer on the surface of graphene prepared by the template method and to study the properties of the synthesized material. 

## 2. Methods of Investigation

### 2.1. Graphene Synthesis

Magnesium oxide (Vekton, Russia, “pure for analysis” grade) was subjected to carbonization with 1,3-bitadiene at 600 °C in a quartz flow reactor. After this reaction the MgO particles were covered with a thin carbon film. Then, the carbon film was cleaned from the MgO template by treatment in a hydrochloric acid solution (Figure 1).

The synthesized graphene was used for doping with boron. 

### 2.2. B-Carbon Nanomaterial Synthesis

Phenylboronic acid (C_6_H_5_-B-(OH)_2_) (Aldrich, USA) was used as a boron-containing precursor. Its structure containing a phenyl group favored the introduction of boron into the synthesized graphene framework. 

Graphene prepared using the template method was treated with phenylboronic acid dissolved in the mixture containing 80% acetone and 20% ethanol. Then, it was placed into an autoclave as described earlier [28]. The autoclave was heated to 650 °C and kept at this temperature for 2 h. A black carbon powder was obtained after the carbonization of the graphene precursor. The amount of carbon doped with boron deposited on graphene was controlled by the change of the sample weight. The synthesized B-carbon nanomaterial was studied using various physical methods. 

### 2.3. Physical Methods for Investigation of B-Carbon Nanomaterial

High-resolution transmission electron microscopy (HRTEM) using a ThemisZ (Thermo Fisher Scientific, USA) electron microscope with an accelerating voltage of 200 kV and maximum lattice resolution of 0.07 nm was used for the investigation of the structure and microstructure of the catalysts. The TEM images were recorded with a Ceta 16 (Thermo Fisher Scientific, USA) CCD matrix. Elemental analysis was performed with Super-X EDS detector (Thermo Fisher Scientific, USA). High-angle annular dark-field images (HAADF STEM image) were recorded using a standard ThemisZ detector. The samples for the HRTEM study were deposited on a holey carbon film mounted on a copper grid by the ultrasonic dispersal of the B-graphene suspension in ethanol.

X-ray photoelectron spectra were recorded on a SPECS (Germany) photoelectron spectrometer using a hemispherical PHOIBOS-150-MCD-9 analyzer (Mg K_α_ radiation, hν = 1253.6 eV, 150 W). The binding energy (BE) scale was pre-calibrated using the positions of the peaks of Au4f_7/2_ (BE = 84.0 eV) and Cu2p_3/2_ (BE = 932.67 eV) core levels. The samples in the form of powder were loaded onto a conducting double-sided copper scotch. The survey and the narrow spectra were registered at the analyzer pass energy 20 eV. Atomic ratios of the elements were calculated from the integral photoelectron peak intensities, which were corrected by corresponding sensitivity factors based on Scofield photoionization cross-sections [29].

Raman spectra were measured using a Horiba Jobin Yvon Lab-Ram HR spectrometer. Excitation was supplied by an argon ion laser (λ = 488 nm) with 2 cm^−1^ spectral resolution.

The surface areas and porosity of the synthesized graphene and B-carbon nanomaterial were determined using an ASAP-2400 (Micromeritics, Norcross, GA, USA) instrument [30].

The boron concentration in B-carbon nanomaterial was determined by the atomic emission spectroscopy. 

## 3. Results and Discussion

### 3.1. HRTEM Study of Graphene

Graphene synthesized by the template method was studied by electron microscopy. A HRTEM image of graphene is shown in Figure 2a. 

The graphene aggregates have dimensions of about 1 μm (Figure 2a). The size of the carbon globules is determined by the geometrical dimensions of the MgO particles. The thickness of the graphene layer in the synthesized graphene sample was 2–4 layers. The graphene surface area was about 1300 m^2^/g. 

### 3.2. B-Carbon Nanomaterial 

Graphene was doped with boron atoms by heating in an autoclave in the presence of phenylboronic acid, acetone, and ethanol at 650 °C. After doping with boron, the mass of the graphene sample increased by 70%. 

### 3.3. B-Carbon Nanomaterial Study by Electron Microscopy

HRTEM images of B-carbon nanomaterial are shown in Figure 2b,c. It also has globular structure (Figure 3b), and additional graphene carbonization led to an increasing of the graphene layer thickness to 3–8 layers.

Figure 3 presents an HAADF STEM image of B-carbon nanomaterial (a) and EDX mapping of boron, carbon, and oxygen atoms (b).

The boron concentration in the B-carbon nanomaterial sample determined by EDX was equal to 4 wt.%. This boron concentration is in agreement with the results of the elemental analysis by atomic emission spectroscopy. According to the latter, the content of atomic boron in B-carbon nanomaterial was 4.3 wt.%.

### 3.4. Investigation of the B-Carbon Nanomaterial Pore Structure

The nitrogen adsorption-desorption isotherms measured on pristine graphene and B- carbon nanomaterial are presented in Figure 4. 

Specific surface area of the synthesized B-carbon nanomaterial was equal to 800 m^2^/g. The deposition of the additional layer of graphene doped with boron led to the surface area decreasing from 1300 to 800 m^2^/g. 

The pore size distributions of pristine graphene and of the B-carbon nanomaterial sample are presented in Figure 5. 

The data presented in Figure 4 and Figure 5 demonstrate that the deposition of an additional boron-containing carbon layer on the surface of graphene leads to the decrease of both the volume and diameter of the pores. 

B-carbon nanomaterial takes the shape of hollow spheres. According to the pore size distribution (Figure 5), its main pores are in the range of 5–25 nm. Note that the diameter of the graphene “spheres” approximately matches the size of the MgO template particles (5–20 nm), which were used for the synthesis of graphene. 

### 3.5. B-Carbon Nanomaterial Study by Raman Spectroscopy

B-carbon nanomaterial samples were studied by Raman spectroscopy to determine specific features of their structure. The first-order Raman spectra of graphene and B-carbon nanomaterial are shown in Figure 6. The obtained results are also presented in Table 1. The data obtained for B-carbon nanomaterial were compared those determined for the graphene sample carbonized in the autoclave with the acetone-ethanol mixture at 650 °C in the absence of phenylboronic acid. 

Two typical modes were observed in the Raman spectra of the synthesized carbon nanomaterial samples: G mode (1597 cm^−1^), related to vibrations of carbon atoms in the graphene layer, and D mode (1355 cm^−1^), related to the presence of carbon atoms in sp^3^ hybridization [31]. The appearance of the defect band in the spectrum of carbon materials is associated with the disordering of the graphite structure. The D/G ratio is commonly used to estimate the defectiveness of carbon nanomaterials. In the case of the studied graphene sample, the ratio is D/G=1.74 (Figure 6 and Table 1). Additional information on the graphene structure can be obtained from smaller peaks identified in the Raman spectra: G′ (≈2700 cm^−1^), D + D′ (≈2950 cm^−1^). The Raman spectrum confirmed that the synthesized carbon material had a graphene-like local structure. Therefore, according to the Raman data, B-carbon nanomaterial is a defective graphene-like carbon nanomaterial.

The data presented in Table 1 demonstrate that the introduction of boron into the graphene structure results in an increase of the D/G ratio from 1.74 to 1.81, i.e., the concentration of defects substantially increases. 

Based on the results of B-carbon nanomaterial characterization by physical methods, the following general scheme of its synthesis can be suggested (Figure 7).

The suggested method includes the graphene synthesis using the template method, followed by deposition of an additional graphene layer doped with boron in an autoclave at 650 °C using a mixture of phenylboronic acid, acetone, and ethanol. After this carbonization procedure, the mass of the graphene sample increased by 70%. The boron concentration in B-carbon nanomaterial determined by different physical methods was about 4 wt.%. The deposition of an additional graphene layer doped with boron led to an increase of the graphene layer thickness from 2–4 to 3–8 monolayers, and a decrease of the specific surface area from 1300 to 800 m^2^/g. 

### 3.6. XPS Study of B-Carbon Nanomaterial

The state of carbon and boron in B-carbon nanomaterial samples was studied by X-ray photoelectron spectroscopy. Survey scans of pristine and B-carbon nanomaterial samples are presented in Figure 8.

Figure 8 demonstrates that the initial graphene contains 2% O and 98% C, making it is a high-quality nanomaterial. The oxygen concentration in B-carbon nanomaterial increases to 6% in addition to the appearance of 4% B. The observed changes are related to the incorporation of boron-oxygen fragments into the graphene structure. 

The C1s spectrum of the B-carbon nanomaterial sample is presented in Figure 9. The high-resolution C1s spectrum can be divided into five individual peaks (Figure 9), respectively referring to C–B (283.6 eV) [32], C=C (284.5 eV), C–OH, C–O–C (285.8 eV), C=O (287.2 eV) and O–C=O (289.1 eV) functional groups [33,34,35]. The satellite peak at the binding energy of about 291 eV attributed to the π-π* transition.

The concentrations of different elements in B-carbon nanomaterial were determined by XPS. The results are presented in Table 2. 

A more accurate conclusion on the state of boron can be made from the binding energy for the B1s peak of the B-carbon nanomaterial. The B1s peak of B-graphene sample is shown in Figure 10. According to the literature data, the binding energy of 193.1 eV is typical for the C-BO_2_ bond [19,36,37]. 

Note that the binding energy of the B1s peak for the C–B bond is in the range of 188–189 eV. The shift of the B1s peak to higher binding energies indicates that the boron atom is bonded with an oxygen atom. So, the following model structure can be suggested for the C–BO_2_ fragment in B-carbon nanomaterial (Figure 11).

We believe that the incorporation of boron atoms into a formed carbon nanomaterial is difficult because boron atoms are larger that carbon atoms. It was shown [38] that the C–C distance in graphene is equal to 1.41 Å, whereas the C–B distance should be about 1.48 Å. Furthermore, boron atoms are firmly bound to oxygen. 

The proposed structure of a fragment of the boron-doped carbon layer deposited on the surface of the initial graphene is shown in Figure 11. This layer is formed during the doping of the initial graphene. During this process the fragments are incorporated into the hexagonal lattice of the carbon monolayer. Therefore, there are two layers in Figure 7: a layer of the initial graphene and a layer of boron-doped graphene. 

## 4. Conclusions

A new method for the synthesis of B-carbon nanomaterials has been developed. The method includes graphene synthesis using the template method, followed by the deposition of an additional layer of boron-doped graphene in an autoclave at 650 °C using a mixture of phenylboronic acid, acetone, and ethanol. Following the graphene carbonization in the autoclave, its weight increased by 70%. The presence of boron atoms in the graphene framework was confirmed by atomic emission spectroscopy, XPS, and EDX. The boron concentration in B- carbon nanomaterials was found to be about 4 wt.%. The deposition of an additional graphene layer doped with boron led to an increase of the graphene layer thickness from 2–4 to 3–8 monolayers, and a decrease of the surface area from 1300 to 800 m^2^/g. The Raman spectrum confirmed that the synthesized carbon material had a graphene-like local structure. According to the XPS data, boron in B-carbon nanomaterials is present in the form of C–BO_2_ fragments. 

## Figures and Tables

**Figure 1 materials-16-01986-f001:**
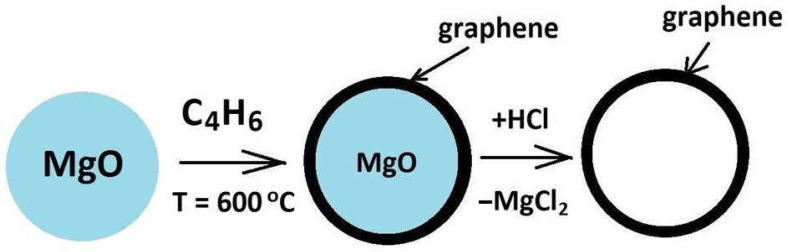
Scheme of graphene synthesis.

**Figure 2 materials-16-01986-f002:**
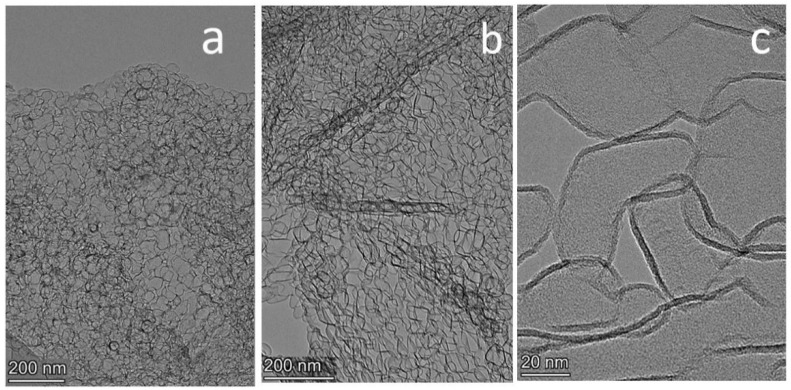
HRTEM images of graphene (**a**) and B-carbon nanomaterial at different magnifications (**b**,**c**).

**Figure 3 materials-16-01986-f003:**
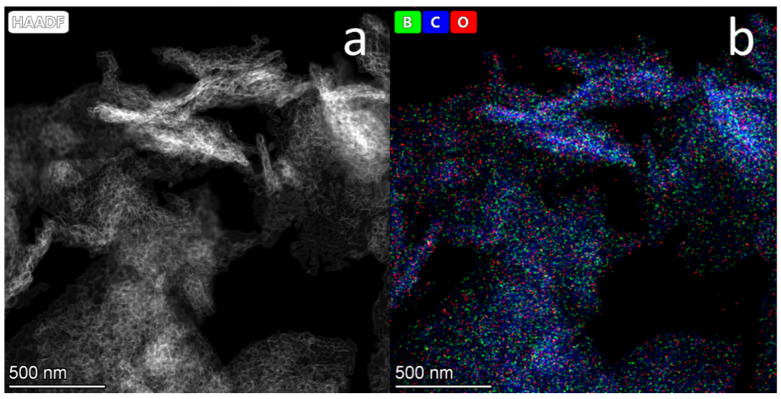
HAADF STEM image of B-carbon nanomaterial (**a**) and EDX mapping of boron, carbon and oxygen (**b**).

**Figure 4 materials-16-01986-f004:**
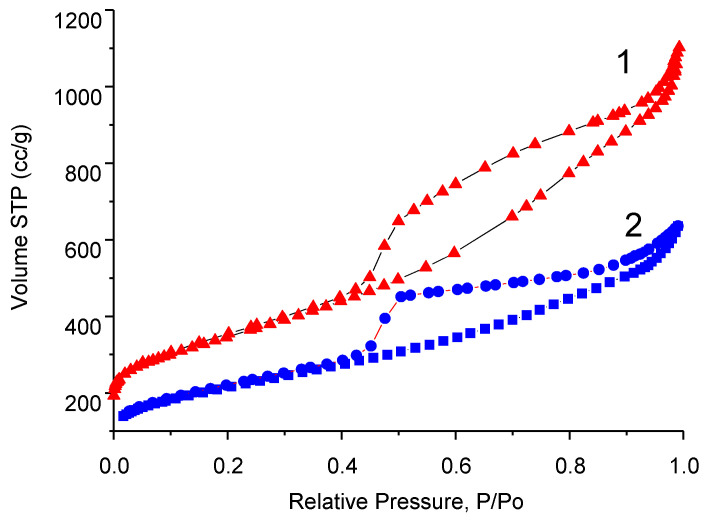
Nitrogen adsorption–desorption isotherms of pristine graphene (1) and B-carbon nanomaterial (2).

**Figure 5 materials-16-01986-f005:**
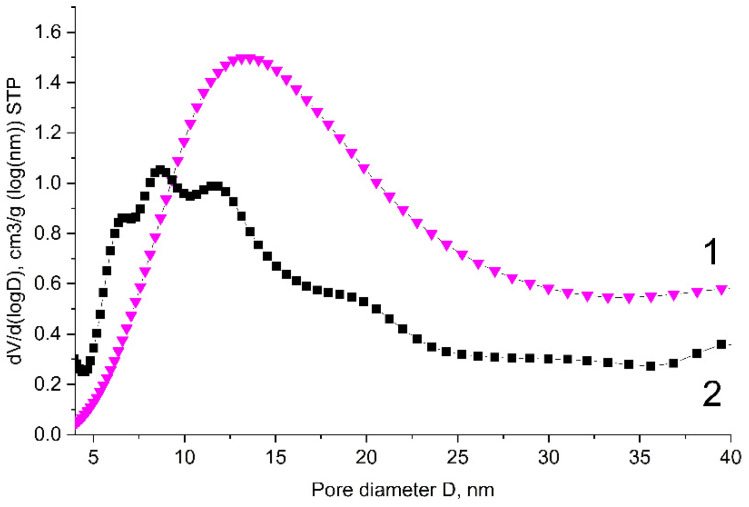
Pore size distributions of pristine graphene (1) and B-carbon nanomaterial (2) (cylindrical pores, QSDFT adsorption branch).

**Figure 6 materials-16-01986-f006:**
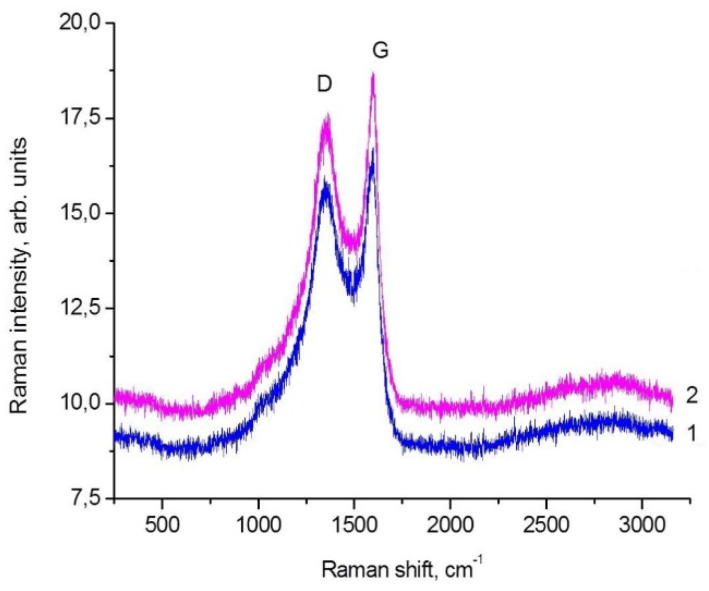
Raman spectra of graphene (1) and B-carbon nanomaterial (2) samples.

**Figure 7 materials-16-01986-f007:**
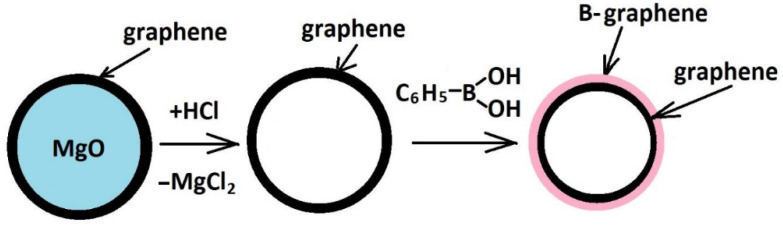
General scheme of B-carbon nanomaterial synthesis.

**Figure 8 materials-16-01986-f008:**
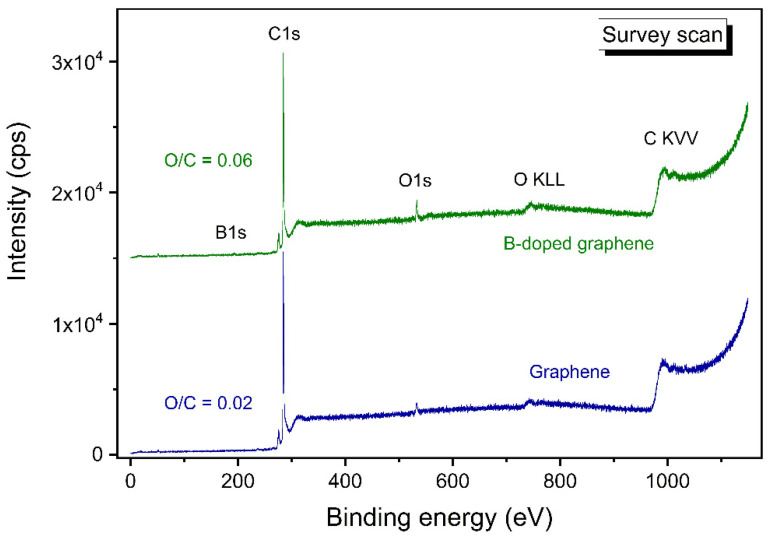
Survey scans of pristine and B-carbon nanomaterial samples.

**Figure 9 materials-16-01986-f009:**
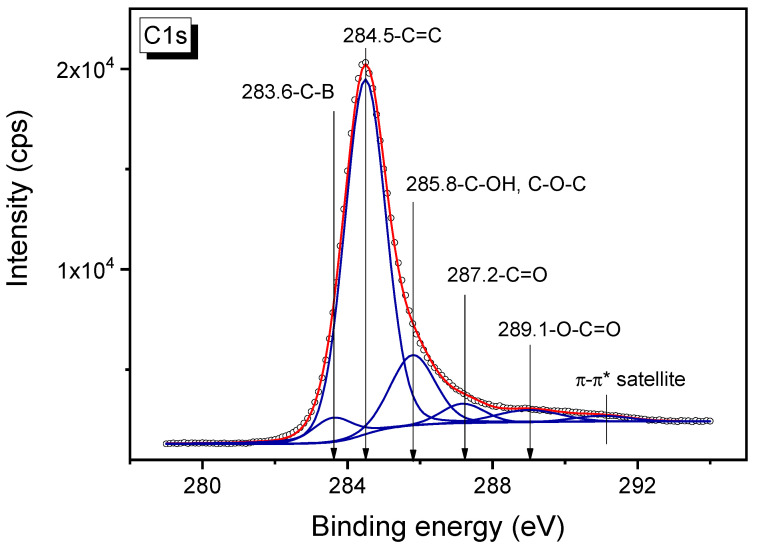
C1s XPS spectrum of B-carbon nanomaterial.

**Figure 10 materials-16-01986-f010:**
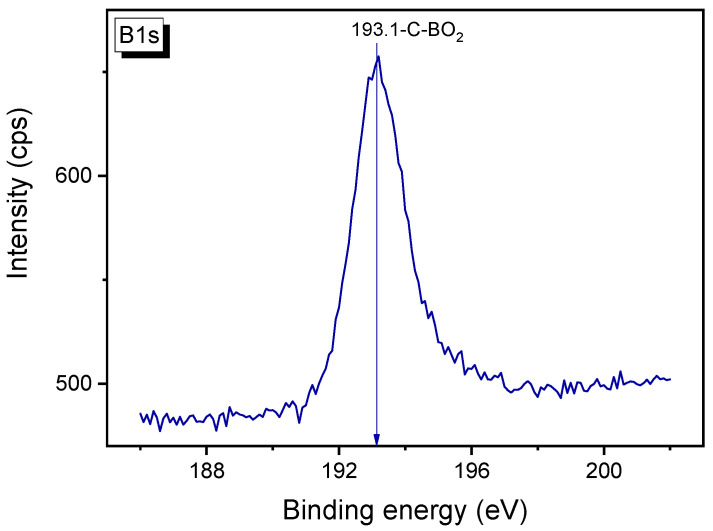
B1s XPS spectrum of B-carbon nanomaterial.

**Figure 11 materials-16-01986-f011:**
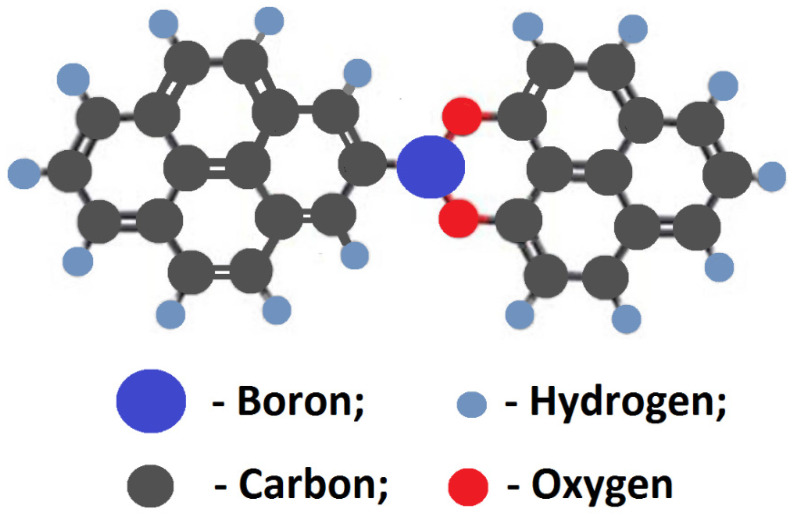
Model structure of the C-BO_2_ fragment in B-carbon nanomaterial.

**Table 1 materials-16-01986-t001:** Raman parameters of graphene and B-carbon nanomaterial samples.

Sample	Frequency, cm^−1^		D/G
Graphene	1355	1597	1.74
B-carbon nanomaterial	1355	1597	1.81

**Table 2 materials-16-01986-t002:** Elemental analysis of the B-carbon nanomaterial sample by XPS.

Element	Concentration, wt.%
Boron	4
Carbon	90
Oxygen	6

## Data Availability

Not applicable.

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
