# Peer review of "Synthesis of Boron-Doped Carbon Nanomaterial"

_materials, 2023, doi:10.3390/ma16051986_

Round 1

Reviewer 1 Report

Present work is on the preparation of boron-doped graphene and its characterization.

The theme is interesting but the  analyses are insufficient to back the authors' claim.

i) Atomic resolution images of graphene and B-doped graphene samples are required.

ii) Please explain why weight increase in B-doped sample has occured

iii) The EDS instrument is not shown in the experimental section.

iv) Figs4 and 5 must be shown with measurements of pristine graphene sample.

v)Wide XPS can be shown for pristine and doped samples so that we can see what elements are in the samples. It will show whether the sample is graphene ore oxidized graphene.

vi) Fig7 is not backed from the results shown in the paper. If Fig10 is the proposed fragment that is deposited on pristine graphene, this work is not about boron doping.

Reviewer 2 Report

The authors report on the synthesis of boron-doped graphitic materials through the MgO templated CVD process followed by treatment with phenylboronic acid at a high temperature. The synthesis of graphitic or graphene-based materials through the MgO templated CVD method has been widely reported. The doping of boron with high-temperature phenylboronic acid has also been widely reported. In general, there is nothing new or novel for the present work. Besides, it is not appropriate to name the obtained materials as graphene directly. As a result, I do not recommend the accpetance of the manuscript.   

Reviewer 3 Report

Dear Editor,

There are almost sufficient works have been done in this paper with the title of "Synthesis of Boron-Doped Graphene" to achieve a useful product which in my view it is good to have accepted in the journal of materials. However, there are some parts that needed to be corrected which are sorted below:

1. The abstract is incomplete, it should contain all of the detailed information about the prominent achievements of the experiments include numeric results.

2. In the introduction, goals should be mentioned in the latest paragraph but the authors report on the work process. It needs to be changed.

3. In the experimental section, it is better to depict any test by a flowchart or simple schematics.

4. In the conclusion, the writer should bring their numeric results with their discussions altogether.

5. Unfortunately, I could not find any novelty in this work. Please elucidate it.

Reviewer 4 Report

The authors report an interesting study concerning a new method of synthesis of boron-doped graphene. This study provides additional insight into the composition and internal microstructure of the boron-doped material, as well as the specifics of the structural ordering of boron atoms in the graphene layer. The article is complete, well organized, and properly referenced. Experimental data are presented and discussed in an understandable manner. The analysis is consistent and well supported by data and explanations. Thus, this manuscript is worthy of publication.

Author Response

We thank the reviewer for his interest in our work and for its high appreciation.

Round 2

Reviewer 1 Report

As the authors state themselves,  it is very difficult to dope boron in graphene network. Moreover, if the graphene is of high quality, it is even harder to dope boron even a few percent. Therefore, in the absence of atomic resolution image, the results only shows that there are some boron compound attached or deposited at the surface which is not doping.

Your instrument can be set to 60 or 80 keV and you will have less knock-on damage. If contamination occurs, it can also indicate that the supposed boron doped layer is pretty 'dirty'.

Author Response

Response to Reviewer #1:

  1. i) As the authors state themselves, it is very difficult to dope boron in graphene network. Moreover, if the graphene is of high quality, it is even harder to dope boron even a few percent. Therefore, in the absence of atomic resolution image, the results only shows that there are some boron compound attached or deposited at the surface which is not doping.

Answer

The main statement of the authors is as follows. It is very difficult to dope boron in graphene network.  But never the less it is possible, which is confirmed in the work [19].

  1. Sheng, Z.; Gao, H.; Bao, W.; Wang, F.; Xia, X. Synthesis of boron doped graphene for oxygen reduction reaction in fuel cells, J. Mater. Chem., 2012, 22, 390–395 DOI: 10.1039/c1jm14694g

In this work [19], boron-doped graphene was synthesized by thermal treatment of graphene oxide in the presence of boron oxide (B2O3).

Our claim is that boron doping easily occurs when forming a graphene network from components of the reaction medium (phenylboronic acid, acetone and ethanol) at 650 ° C. For example, ethanol at elevated temperatures easily loses water to form ethylene. Carbon is formed from ethylene at a high rate. Separate experiments have shown that a mixture of acetone and ethanol (without phenylboronic acid) produces carbon on both the surface of magnesium oxide and graphene. 

Electron microscopy data have shown that there are no new phases or nanoparticles on the surface of the graphene layer. Boron is distributed evenly over the surface of graphene. XPS data clearly indicate that boron is introduced into the carbon structure in the form of fragments of C-B-O2.  

ii) Your instrument can be set to 60 or 80 keV and you will have less knock-on damage. If contamination occurs, it can also indicate that the supposed boron doped layer is pretty 'dirty'.

Answer

It should be noted that the obtained electron microscopic images already have atomic resolution.

No contamination of the boron-doped carbon layer occurs. The problem is the displacement of graphene under the beam, which leads to low-quality images.

Rebuilding an expensive instrument to 60 or 80 keV is difficult for technical reasons.

Reviewer 2 Report

I do not think the manuscript has enough novelty to support its publication. Besides, Figure 5 and 6 showing reduced pore size and surface area can not demonstrate the conclusion that deposition of an additional boroncontaining carbon layer on the surface of graphene.  

Author Response

Response to Reviewer #2:

  1. Besides, Figure 5 and 6 showing reduced pore size and surface area cannot demonstrate the conclusion that deposition of an additional boron containing carbon layer on the surface of graphene.  

 Answer

The reviewer is right that only from the measurement of the porous structure (Figure 5 and 6) it is impossible to make the conclusion that deposition of an additional boron containing carbon layer on the surface of graphene. However, this conclusion is drawn from a set of physical methods. Electron microscopic images unequivocally prove that no additional carbon formations occur. Only an increase the thickness of the carbon layer is observed. This is well consistent with the adsorption data.

  1. I do not think the manuscript has enough novelty to support its publication.

Answer

A new method for synthesis of B-graphene-like carbon nanomaterial has been developed. The method includes graphene synthesis by the template method followed by deposition of an additional layer of boron-doped graphene in an autoclave at 650 °C using a mixture of phenylboronic acid, acetone and ethanol.

The authors will be grateful to the reviewer if he provides links, i.e. articles in which such a method has already been developed.
